# Harnessing nitroarenes as nitrogen and oxygen sources for general oxo-aminomethylation of alkenes

Ting Zhou[1], Jing Nie [1], Chi Wai Cheung [1,2] ✉ & Jun-An Ma [1] ✉

Amino alcohols are essential synthetic building blocks and privileged motifs in drug development, playing a crucial role in modulating pharmacokinetics and biological activity. However, the efficient synthesis of 3-amino alcohols remains challenging compared to their 2-amino counterparts, often requiring multistep procedures, expensive substrates, or highly sensitive reagents. Despite their versatile reactivity in advanced synthesis, nitroarenes remain underutilized as dual nitrogen and oxygen sources. Herein, we report a metallaphotoredox-catalyzed multicomponent oxo-aminomethylation of nitroarenes, tertiary alkylamines, and alkenes, providing a modular and cost-effective route to diverse 3-arylamino alcohols. This strategy features a broad substrate scope, excellent functional group tolerance, and high regioselectivity. The resulting amino alcohols serve as key intermediates for further derivatization, enhancing molecular complexity. By expanding the synthetic utility of nitroarenes, this method offers a practical and efficient pathway to bioactive molecules with pharmaceutical relevance.

Amino alcohols are privileged structural motifs in drug development due to their combination of polar amino and hydroxyl groups[1–3], which enable unique biological interactions and enhance the pharmacokinetic properties of drug candidates. Beyond being emerging components in approved drugs[4,5], amino alcohols play a crucial role in asymmetric synthesis[6–8] and materials science[9]. Consequently, the synthesis of amino alcohols with diverse architectures and functional decorations remains a prominent research focus in the synthetic chemistry community.

While numerous structurally diverse 2-amino alcohols have been extensively explored[10–18], 3-amino alcohols remain less investigated[19], likely due to synthetic challenges and the limited availability of suitable chemical feedstocks. This has constrained the incorporation of 3-amino alcohol units in organic synthesis and drug discovery. Among the available 3-amino alcohols, α-substituted 3-arylamino alcohols—bearing functional groups such as aromatic[20], heterocyclic[21,22], ester[23,24], trifluoromethyl[25–27], and phosphonate moieties[28]—exhibit distinctive physicochemical properties that enhance their potential for

treating various diseases (Fig. 1). However, the synthesis of these structurally complex compounds typically necessitates multistep de novo methods or the use of expensive substrates, hindering the exploration of diverse 3-arylamino alcohol scaffolds for structure-activity relationship studies and novel drug development. Although conventional synthetic routes, such as hydroamination of alkenes[29–31], nucleophilic substitution[32], nitrene insertion into cyclopropyl alcohols[33], and other nucleophilic addition reactions[34], are available (Fig. 2A), these methods are often limited by narrow substrate scope, low step economy, and poor compatibility with organometallic reagents. Therefore, there is a compelling need for modular synthetic strategies utilizing simpler building blocks, enabling efficient and flexible construction of complex 3-arylamino alcohol compounds suitable for bioactive molecule synthesis.

Nitroarenes represent a particularly attractive class of reactants in modern organic synthesis due to their accessibility, low cost, and stability[35]. Recent advances have highlighted the novel reactivity of nitroarenes in combination with alkenes[36,37], as demonstrated by

[1]Department of Chemistry, State Key Laboratory of Synthetic Biology, Tianjin University, Tianjin, PR China. [2]State Key Laboratory of Synthetic Chemistry and Department of Chemistry, The Chinese University of Hong Kong, Shatin, New Territories, Hong Kong, PR China. ✉e-mail: cw.cheung@cuhk.edu.hk; majun_an68@tju.edu.cn

**Fig. 1 | 3-Arylamino alcohols as key structural components in drug development.** *n*Bu *n*-butyl, Me methyl, Et ethyl.

Baran[38], Leonari[39,40], Parasam[41,42], Studer[43], and others[44,45]. Transition metal-catalyzed or photoinduced transformations involving nitroarenes have enabled the synthesis of *N*-alkyl anilines via hydroamination[38,44] (Fig. 2B (i)), alcohols via hydration[43] or carbohydroxylation[45] (Fig. 2B (ii)), 1,2-diols via dihydroxylation[40] (Fig. 2B (iii)), and carbonyl compounds via oxidative cleavage[39,41,42] (Fig. 2B (iv)), with nitroarenes acting as surrogates for anilines or as oxygen atom donors. Despite these developments, the incorporation of both nitrogen and oxygen atoms to form more functionalized compounds, such as 3-arylamino alcohols **1**, remains relatively underexplored (Fig. 2B (v)). These reaction pathways offer a more atom- and step-economical approach to utilizing nitroarenes, providing an effective strategy for building complex, functionalized molecules.

Alkenes, with their broad structural diversity and high reactivity, serve as versatile building blocks for the synthesis of amino alcohols. The simultaneous incorporation of amino, methylene, and hydroxyl subunits into alkenes represents a modular and expedited strategy for constructing 3-amino alcohols. In this context, the List's group reported the cycloaddition of *N*-hydroxymethyl amides **2** with alkenes to generate dihydrooxazines **3**[46] (Fig. 2C (i)). Additionally, they demonstrated the aza-Prins reaction, which affords oxazinanes **4**[47] (Fig. 2C (ii)). These azaheterocycles can be subsequently reduced to yield *N*-alkylated 3-amino alcohols **5**, broadening the synthetic strategies available for amino alcohol construction. More recently, Glorius and colleagues synthesized a bifunctional reagent **6** that enabled the photoinduced amino-silyloxymethylation of alkenes, producing 3-imino silyl ethers **7**[48] (Fig. 2C (iii)). These intermediates undergo reduction or hydrolysis to afford *N*-alkylated or unprotected amino alcohols (**8**). Notably, while preparing our work, Zhang's group reported a reductive three-component synthesis of a subclass of 3-arylamino alcohols—specifically, γ-arylamino-α-hydroxybutyric acids **10**—via the reaction of nitroarenes, formaldehyde, and acrylic acid or esters[49] (Fig. 2C (iv)). This transformation proceeds through a direct or cyclization (via species **9**)/reduction sequence, enabled by their unique cobalt-based heterogeneous catalyst. Despite these advances, challenges such as limited substrate scope, constrained structural diversity, and suboptimal synthetic efficiency highlight the need for more general and streamlined methodologies. The development of such approaches would significantly expand the accessibility of 3-amino alcohols and their derivatives, enhancing their utility in organic synthesis and drug discovery.

Given our current interest in the reductive functionalization of nitroarenes to construct complex aliphatic anilines[50–52], particularly fluorinated amino compounds[52], which are crucial in pharmaceutical[53] and agrochemical[54] science, we present a light-induced synthetic strategy to synthesize 3-arylamino alcohols (Fig. 2D). This approach utilizes readily accessible and commercially available substrates, including nitroarenes, tertiary alkyl amines, and alkenes. The selective and efficient assembly of these compounds is facilitated by a metallaphotocatalytic method, employing a simple photocatalyst and transition metal catalyst. In this reaction, nitroarenes act as the formal donors of arylamino and hydroxyl radicals, while tertiary alkyl amines and alkenes serve as sources of formal methylene carbene and vicinal dicarboradicals, respectively. This modular three-component reaction enables the synthesis of a diverse range of skeletally intricate 3-arylamino alcohols, owing to the successful incorporation of a broad scope of both tertiary alkyl amines and alkenes. Particularly, the method shows excellent tolerance toward a variety of functional groups embedded in alkene partners, including fluorinated, aromatic, heterocyclic, acrylate, polyunsaturated, and heteroatom groups, significantly enriching the molecular complexity and functionality of the resulting 3-arylamino alcohols. Furthermore, versatile transformations at both the amino and hydroxyl groups are achievable. This photocatalytic amination protocol offers a straightforward and expedited route to create structurally complex and highly functionalized 3-arylamino alcohols and derivatives (**11**–**110**), advancing the development of novel bioactive candidates for drug and agrochemical discovery.

## Results and discussion
### Reaction optimization
At the outset, we optimized the three-component oxo-aminomethylation reaction (Table 1; Table S1, Supplementary Information). Building on our previously established reaction conditions for dual nickel/photoredox multicomponent trifluoroalkyl aniline synthesis[52], we employed 4-nitroanisole (**N1**), *N*,*N*-dimethylcyclohexylamine (**A1**, 6.0 equiv.), and 3,3,3-trifluoropropene (**O1**, 1 atm) as the starting materials. Under blue light irradiation for 18 hours, the reaction was conducted with 4CzIPN (**PC1**, 5 mol %) as the photosensitizer, and a combination of nickel(II) nitrate and bathophenanthroline ligand (**L1**, 20 mol %) as the metal catalyst. Hantzsch ester (**HE**, 2.0 equiv.) was employed as the reducing agent, with *N*-methylpyrrolidone (NMP) serving as the solvent. This setup successfully yielded the trifluoromethylated 3-arylamino alcohol, 1,1,1-trifluoro-4-((4-methoxyphenyl)amino)butan-2-ol **11**, in 56% yield (Entry 1). Further screening of ligands, nickel salts, and photocatalysts revealed that the combination of [bipyridine] nickel(II) dichloride complex (Ni(bipy)Cl₂) and the inexpensive 4CzPN photocatalyst (**PC4**) provided the best results, producing the desired

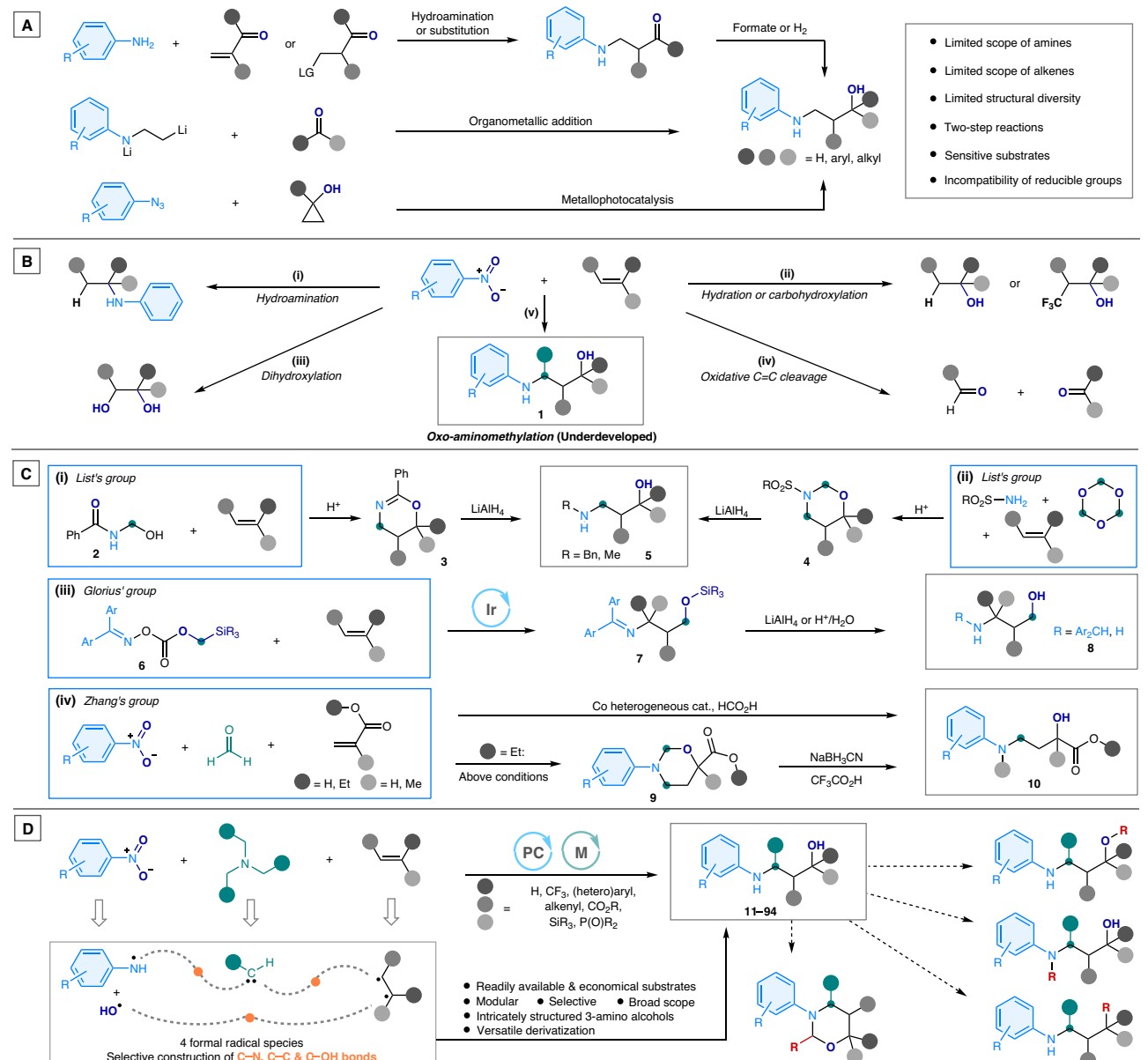

**Fig. 2 | Development of the Construction of 3-Arylamino Alcohols. A** Common synthetic methods for the preparation of 3-arylamino alcohols. **B** Nitroarenes as versatile nitrogen and oxygen sources for alkene functionalization. **C** Recent advancements in alkene transformations to construct 3-amino alcohols. **D** Our development of a metallaphotocatalytic general synthesis of 3-arylamino alcohols using nitroarenes, alkenes and tertiary alkylamines. Ph phenyl, Bn benzyl, Me methyl, Et Ethyl, Ar aryl, PC photocatalyst, M transition metal.

compound in 70% yield (Entries 2–8). By reducing the photocatalyst loading to 3 mol % (Entries 9 and 10), decreasing the amount of Hantzsch ester to 1.8 equiv (Entry 13), and extending the reaction time to 24 hours (Entry 14), the yield was further improved to 78%. Since the use of lower loadings of the Ni(bipy)Cl₂ led to diminished product yields (Entries 11 and 12), 20 mol% of the Ni catalyst was employed for further studies. Control experiments confirmed that the nickel complex was essential for optimal reaction performance (Entry 15), while the addition of Hantzsch ester could further improve the product yield (Entry 16). The optimal conditions were subsequently applied to the scope study of this photocatalytic oxo-aminomethylation reaction (Entry 14).

## Substrate scope

With the optimized conditions established, we investigated the scope of nitroarenes for synthesizing trifluoromethylated 3-arylamino alcohols (Fig. 3). This three-component reaction proved versatile, accommodating a broad range of nitroarenes (**N1–N29**) and nitroheterocycles (**N30–N36**) to yield the corresponding trifluoromethylated 3-arylamino alcohols (**11–46**). A variety of functional groups and drug-related substituents on the nitroaromatic rings were compatible, including alkyl and phenyl ethers (**N1–N5**), *tert*-butyl (**N6**), phenyl (**N7**) and methyl (**N16** and **N17**) groups, thioethers (**N8**), amides (**N9**), fluorides (**N10**), chlorides (**N11**), bromides (**N12**), esters (**N13**), fluoroalkyl ethers (**N14** and **N15**), pinacol boronic esters (**N18** and **N19**), trifluoromethyl groups (**N20**, **N21** and **N26**), aldehydes (**N22**), ketones (**N23**), alkynes (**N24**), and nitriles (**N27**). The position and nature of substituents on the nitroarenes had minimal influence on the reaction, permitting the use of *para-* (**N1**, **N4**–**N16**, **N18**, **N20**, **N22** and **N24**), *meta-* (**N2**, **N19**, **N21**), and *ortho-* (**N3** and **N17**) substituted, as well as mono- (**N1–N24**), di- (**N25–N27**), and trisubstituted (**N28**) nitroarenes. Fused-ring nitroarenes, such as 1-nitronaphthalene (**N29**), also served

**Table 1 | Optimization of the oxo-aminomethylation reaction of 3,3,3-trifluoropropene**

| Entry | PC (mol %) | Ni salt (mol %) and ligand (mol %) | HE (equiv.) | Yield/%[a] |
|---|---|---|---|---|
| 1 | PC1 (5) | Ni(NO$_3$)$_2$·6H$_2$O (20), L1 (20) | 2 | 56 |
| 2 | PC1 (5) | Ni(NO$_3$)$_2$·6H$_2$O (20), L2 (20) | 2 | 31 |
| 3 | PC1 (5) | Ni(NO$_3$)$_2$·6H$_2$O (20), L3 (20) | 2 | 52 |
| 4 | PC1 (5) | Ni(BPhen)Br$_2$ (20) | 2 | 57 |
| 5 | PC1 (5) | Ni(bipy)Cl$_2$ (20) | 2 | 68 |
| 6 | PC2 (5) | Ni(bipy)Cl$_2$ (20) | 2 | 46 |
| 7 | PC3 (5) | Ni(bipy)Cl$_2$ (20) | 2 | Trace |
| 8 | PC4 (5) | Ni(bipy)Cl$_2$ (20) | 2 | 70 |
| 9 | PC4 (4) | Ni(bipy)Cl$_2$ (20) | 2 | 67 |
| 10 | PC4 (3) | Ni(bipy)Cl$_2$ (20) | 2 | 71 |
| 11 | PC4 (3) | Ni(bipy)Cl$_2$ (15) | 2 | 62 |
| 12 | PC4 (3) | Ni(bipy)Cl$_2$ (10) | 2 | 63 |
| 13 | PC4 (3) | Ni(bipy)Cl$_2$ (20) | 1.8 | 74 |
| 14 | PC4 (3) | Ni(bipy)Cl$_2$ (20) | 1.8 | 78[b] |
| 15 | PC4 (3) | Ni(bipy)Cl$_2$ (0) | 1.8 | Trace[b] |
| 16 | PC4 (3) | Ni(bipy)Cl$_2$ (20) | 0 | 68[b] |

Reaction conditions: 4-Nitroanisole (**N1**, 1.0 equiv., 0.10 mmol), N,N-dimethylcyclohexylamine (**A1**, 6.0 equiv.), 3,3,3-trifluoropropene (**O1**, 1atm), photocatalyst (**PC1–PC4**, 3–5 mol %), Ni salt/ligand (0–20 mol %, Ni complex catalyst prepared in situ or as an authentic sample), Hantzsch ester (1.8–2.0 equiv.), NMP (1 mL), –40 °C, blue LEDs (30 W, 455–460 nm), 18 h. Symbols for reactants are shown in bold italics; symbols for photocatalysts and ligands are shown in bold.
[a]Isolated yield.
[b]Reaction time of 24 h.
Me methyl, Et ethyl, Ph phenyl, [t]Bu tert-butyl.

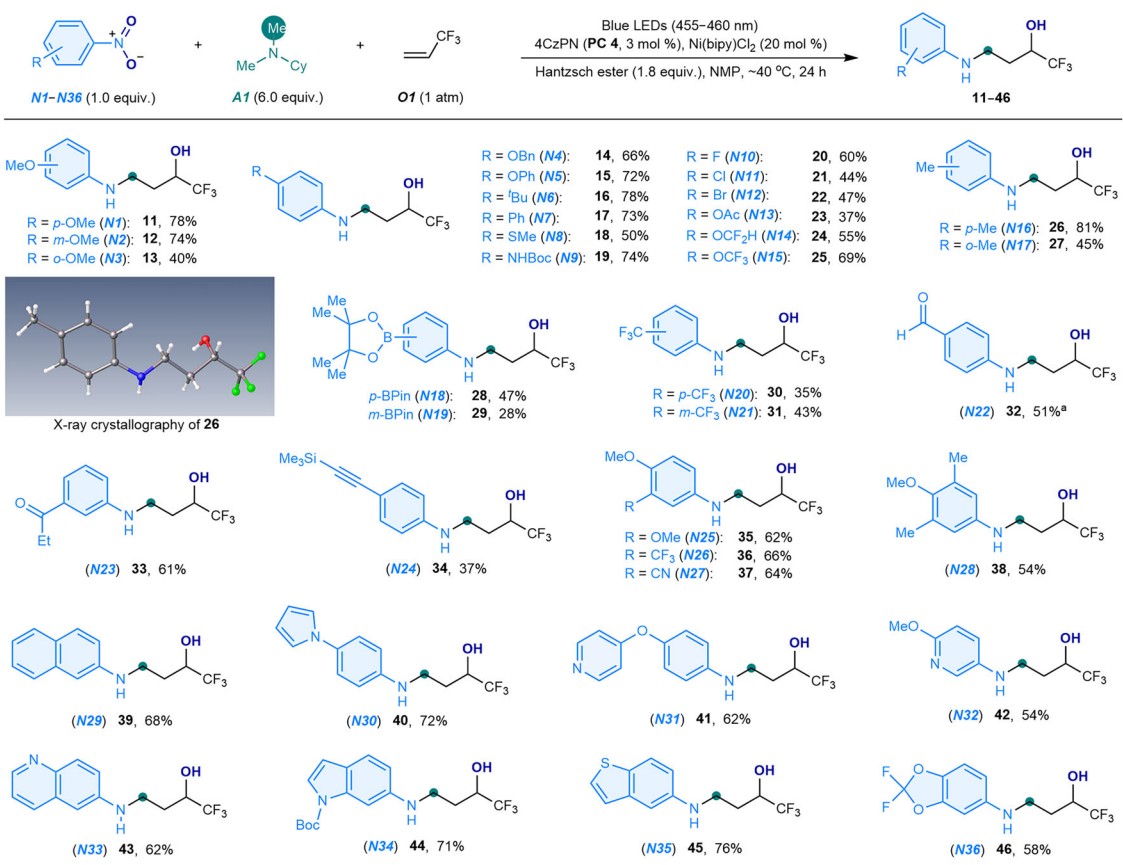

**Fig. 3 | Scope of nitroarenes.** Nitroarene (**N1–N36**, 1.0 equiv., 0.10 mmol), *N,N*-dimethylcyclohexylamine (**A1**, 6.0 equiv.), 3,3,3-trifluoropropene (**O1**, 1 atm), 4CzPN (**PC4**, 3 mol %), Ni(bipy)Cl₂ (20 mol %), Hantzsch ester (1.8 equiv.), NMP (1 mL), -40 °C, blue LEDs (30 W, 455–460 nm), 24 h. Isolated yields are shown.

[a]2-(4-Nitrophenyl)−1,3-dioxolane used as nitroarene substrate; deprotection occurred under the reaction conditions to afford the formyl product. Me methyl, Cy cyclohexyl, *t*Bu *tert*-butyl, Bn benzyl, Ph phenyl, Boc *tert*-butoxycarbonyl, Ac acetyl, BPin pinacol boronate, Et ethyl.

as effective coupling partners. Additionally, a variety of heterocyclic nitroaromatic compounds, including furans (**N30**), pyridines (**N31** and **N32**), quinolines (**N33**), indoles (**N34**), thiophenes (**N35**), and difluorobenzodioxoles (**N36**), were compatible. These *N*-aryl and heteroaryl trifluoromethylated 3-arylamino alcohols represent valuable synthetic scaffolds for the development of fluorinated bioactive molecules. The structures of these trifluoromethyl-based 3-arylamino alcohols were further validated through X-ray crystallographic analysis of compound **26**. Notably, the trifluoroethanol group in these compounds could act as a unique bioisostere, offering significant potential for the design of potent pharmaceuticals[55]. Our reaction method eliminates the need for advanced hydroxytrifluoroethylating agents[55], providing a streamlined strategy for accessing bioactive compounds that incorporate the trifluoroethanol unit.

The versatility of alkenes as building blocks in organic synthesis stems from their adaptability to poly-substitution and functionalization (Fig. 4). Under the optimized reaction conditions, a wide variety of alkenes, including perfluoroalkyl alkenes (**O2** and **O3**), styrenes (**O4–O11**), heteroaryl alkenes (**O12–O15**), dienes (**O16**), and acrylic acid derivatives (**O17–O22**), were efficiently incorporated into the 3-arylamino alcohol products (**47–69**). Styrene substrates tolerated various functional groups on their aromatic rings, such as amides (**O5**), trifluoromethyls (**O6**), boronic esters (**O7**), alkenes (**O8**), and perfluorophenyls (**O9**). Additionally, alkenes containing fused carbocycles and heterocycles, such as naphthylenes (**O10** and **O11**), pyridines (**O12**), thiophenes (**O13** and **O14**), and thiazoles (**O15**), were suitable reaction partners. Regioselective oxo-aminomethylation occurred at the terminal alkene of (*E*)-buta−1,3-dien-1-ylbenzene (**O16**), yielding *trans*-allyl alcohol-adorned aniline. A range of alkyl

acrylates (**O17–O21**) and acrylamides (**O22**) with varying steric bulk reacted smoothly to produce γ-amino-α-hydroxybutyric acid derivatives (**62–69**), which are valuable scaffolds in drug development[49]. Furthermore, 1,1-disubstituted alkenes (**O23–O31**), including those with trifluoromethyl and diverse aryl, heterocyclic, and ester groups, reacted successfully to yield densely functionalized 3-arylamino alcohols (**70–78**). These aniline compounds feature quaternary carbons with four distinct substituents, enhancing stability and site selectivity for drug design[56,57]. Sterically hindered internal alkenes, such as hexafluorobut-2-ene (**O32**) and (cyclopropylidenemethyl) benzene (**O33**), also underwent smooth reaction, producing polyfluorinated (**79**) and cyclopropane-decorated (**80**) 3-aminoaryl alcohols, respectively. Moreover, oxo-aminomethylation of α-silyl- and phosphonyl-substituted alkenes (**O34** and **O35**) was successfully achieved, affording silylated and phosphorylated 3-arylamino alcohols (**81** and **82**). The broad applicability of this modular method for synthesizing 3-arylamino alcohols, which are otherwise challenging to access, underscores its generality and practicality, offering valuable applications in organic synthesis and facilitating structure-activity relationship studies in drug discovery.

Enhancing saturation and three-dimensionality can improve the lipophilicity and specificity of drug molecules, aiding the identification of hit compounds for drug development[56]. This strategy can be achieved by using higher-membered tertiary alkylamines as reaction components (Fig. 5). Under the optimized conditions, tripropylamine (**A5**), tripentylamine (**A6**), triisopentylamine (**A7**), and tris(3,6-dioxaheptyl)amine (**A8**) successfully reacted with 4-nitroanisole and 3,3,3-trifluoropropene, yielding a series of highly branched and complex 3-arylamino alcohols with diverse

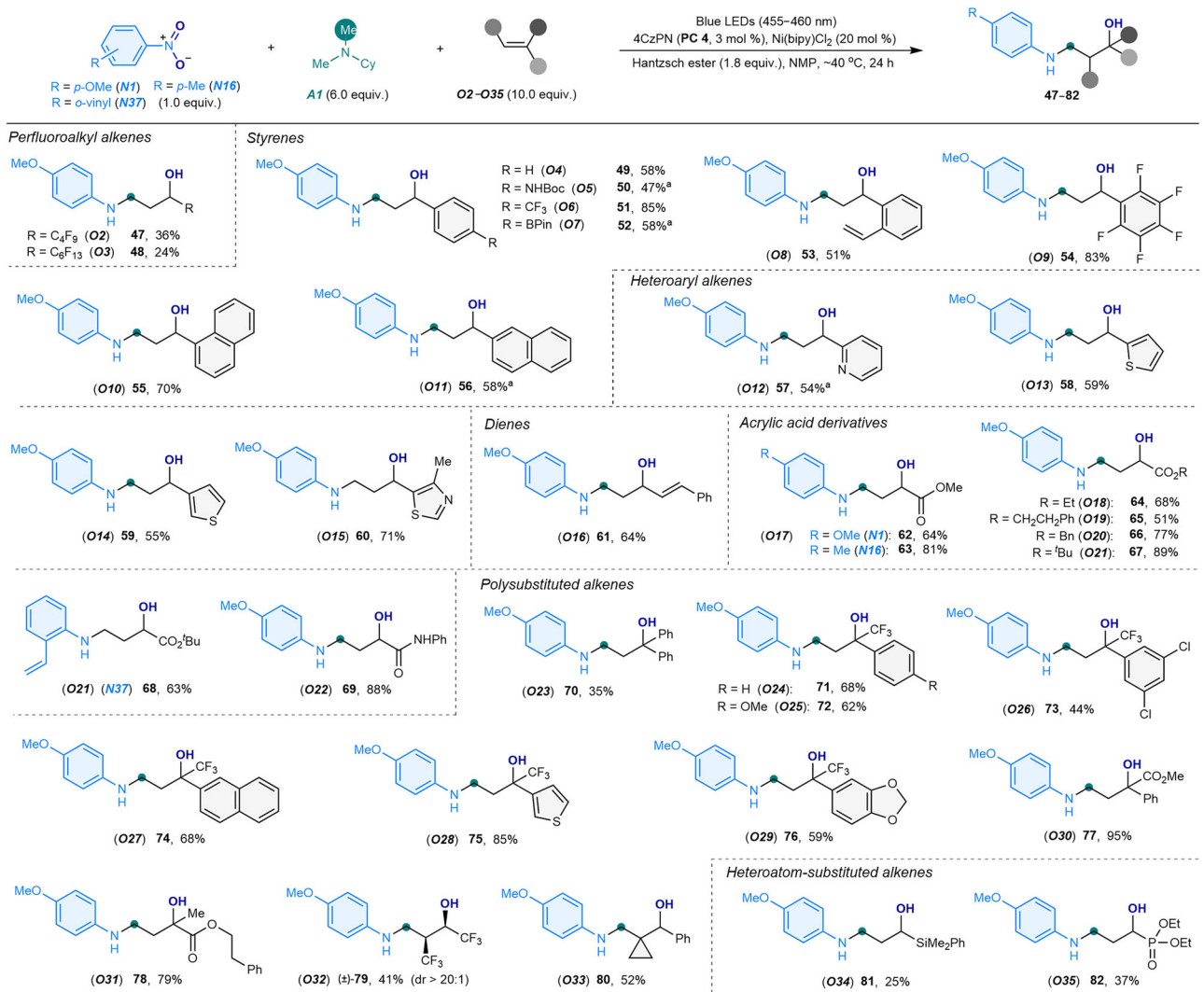

**Fig. 4 | Scope of alkenes.** Nitroarene (**N1** or **N16** or **N37**, 1.0 equiv., 0.10 mmol), *N,N*-dimethylcyclohexylamine (**A1**, 6.0 equiv.), alkene (**O2–O35**, 10.0 equiv., 1.0 mmol), 4CzPN (**PC4**, 3 mol %), Ni(bipy)Cl₂ (20 mol %), Hantzsch ester (1.8 equiv.), NMP (1 mL), -40 °C, blue LEDs (30 W, 455–460 nm), 24 h. Isolated yields are shown. X-ray crystallography of **49** and **71** data were obtained (see Supporting Information for details). ª1,3-Oxazinanes were formed in 23–45% yield as co-products (see supporting Information for details). Me methyl, Cy cyclohexyl, Boc tert-butoxycarbonyl, BPin pinacol boronate, Et ethyl, Ph phenyl, Bn benzyl; ᵗBu *tert*-butyl.

side-chain lengths (**83–86**). The X-ray crystallographic structure of compound **86** revealed that the reaction with higher-membered tertiary alkylamines (**A5–A8**) affords 3-arylamino alcohols with an *anti*-1,3-substituted configuration. Although the yields were modest due to the increased steric bulk of these tertiary alkylamines, these structurally complex amines offer potential for designing intricately functionalized bioactive molecules for further investigation.

Our metallaphotocatalytic oxo-aminomethylation reaction selectively affords 1,3-arylamino alcohols as the sole products from all nitroarenes and most alkenes (Figs. 3–5). Reactions with certain alkenes (**O5**, **O7**, **O11** and **O12**, Fig. 4) yield the corresponding 1,3-oxazinanes in yields ranging from 23% to 45%, likely due to the cyclization of the 1,3-arylamino alcohol products with formaldehyde generated in situ (vide infra). Furthermore, electron-rich alkenes, such as 4-phenylbut-1-ene, vinyl acetate, and ethoxyethene, failed to undergo the reaction to afford the corresponding 1,3-arylamino alcohols. Nevertheless, this reaction generally provides a straightforward and efficient method for synthesizing structurally diverse and functionalized amino alcohols, offering significant potential for advancing organic synthesis and pharmaceutical applications.

## Synthetic utility

The 3-arylamino alcohol compounds not only incorporate the trifluoroethanol moiety, which facilitates novel drug development, but also provide reactive amino and hydroxy groups that serve as handles for multiple chemical transformations. The synthetic utility of this oxo-aminomethylation reaction was systematically explored:

(1) Large-scale synthesis (Fig. 6A). The reaction protocol demonstrates good scalability. Reactions using 5 to 8 mmol of nitroarenes coupled with gaseous 3,3,3-trifluoropropene (**O1**) and liquid styrene (**O4**) afforded the corresponding 3-arylamino alcohols **11** and **49** in 66% and 53% yields, respectively. The productivity of large-scale synthesis is comparable to that achieved on the microgram scale (Figs. 3 and 4).

(2) Synthesis of bioactive molecules (Fig. 6B). Nitroarenes incorporating the ibuprofen scaffold (**N38**), as well as nitrofen (**N39**), reacted smoothly to afford 3-amino alcohol-embedded pharmaceutical and herbicide variants (**87** and **88**). By utilizing inexpensive nitroarenes and styrenes as modular substrates, a variety of 3-arylamino alcohols can be readily synthesized as bioactive molecules (**89–94**), serving as potential inhibitors[20,58], for disease treatments and circumventing the need for traditional multistep synthetic methods.

**Fig. 5 | Scope of tertiary alkylamines.** 4-Nitroanisole (*N1*, 1.0 equiv., 0.10 mmol), tertiary alkylamine (*A5–A8*, 6.0 equiv.), 3,3,3-trifluoropropene (*O1*, 10.0 equiv., 1.0 mmol), 4CzPN (**PC4**, 3 mol %), Ni(bipy)Cl$_2$ (20 mol %), Hantzsch ester (1.8 equiv.), NMP (1 mL), ~40 °C, blue LEDs (30 W, 455–460 nm), 24 h. Isolated yields are shown. Me methyl, Cy cyclohexyl, Et ethyl, $^n$Bu *n*-butyl.

(3) Product derivatization (Fig. 6C). The 3-arylamino alcohols proved to be versatile building blocks for organic synthesis. For example, using the trifluoromethylated 3-arylamino alcohol (**11**) as a starting material, various *N*- and *O*-functionalization were achieved, yielding *N*-methylated (**95**), *O*-benzylated (**97**), *O*-acylated (**99**), *N*-acylated (**100**), and *N*-benzylated (**101** and **102**) analogs. *N*- and *O*-difunctionalization was also fully realized upon the application of additional carbon electrophiles, resulting in the formation of compounds **96** and **98**. Notably, α-phenyl 3-arylamino alcohol (**49**) underwent *O*-arylation to afford the aminotrifluoroalkyl aryl ether (**103**), which is structurally related to the antidepressant drug fluoxetine (bottom right), presenting a promising scaffold for the design of psychotropic drugs. The demethylation and dehydroxylative chlorination of **11** yielded *N*-phenol-substituted amino alcohol (**104**) and chloro-trifluoroalkyl aniline (**105**) derivatives, respectively. Moreover, the use of paraformaldehyde, triphosgene, and phosphoryl chloride as linkers enabled the cyclization of amino alcohols **11** and **67**, resulting in the formation of 1,3-oxazinane (**106**), 1,3-oxazinan-2-one (**107**), and 1,3,2-oxazaphosphinane 2-oxide (**108**) rings.

(4) Product derivatization for structural elucidation (Fig. 6D). To elucidate the stereochemical structures of 3-arylamino propanol products bearing two stereocenters (Figs. 4 and 5), compounds **79** and **83** were subjected to an annulation reaction with triphosgene to afford the corresponding 1,3-oxazinan-2-ones **109** and **110**, respectively, thereby facilitating the growth of single crystals for X-ray crystallographic analysis. The resulting X-ray structures confirmed a *cis*-configuration between the two vincinal CF$_3$ groups in compound **109** and between the 1-ethyl and 3-CF$_3$ groups in compound **110**, thereby corroborating the *syn*- and *anti*-configurations of the parent 3-arylamino propanols **79** and **83**, respectively.

Overall, these amino alcohol derivatives serve as alternative synthetic synthons in organic synthesis and as structural units for the development of drug-related compounds, facilitating the creation of novel functional molecules and effective pharmaceuticals.

## Mechanistic study

To investigate the mechanism of the oxo-aminomethylation reaction, we performed control experiments and conducted instrumental analyses to identify the reacting species involved:

(1) Probing the source of the hydroxyl group. The hydroxyl group in the 3-arylamino alcohol products could originate from water, either as residual moisture in the reaction mixture or as a byproduct of nitroarene reductive deoxygenation. To determine whether water contributes to hydroxyl incorporation, we conducted the oxo-aminylmethylation reaction in the presence of excess $^{18}$O-labeled water (Fig. 7A). However, high-resolution mass spectrometry (HRMS) analysis confirmed that only the unlabeled products **67** and **77** was obtained, with no detectable $^{18}$O incorporation. These results indicated that the oxygen atom from the nitro group of the nitroarenes is likely transferred directly to the alkenes to generate the hydroxyl group in the 3-arylamino alcohols, rather than arising from hydroxylation by water.

(2) Probing the carbocation and ketyl intermediates for product synthesis. We hypothesized that the 3-arylaminopropyl cation species (*Int-1*, Fig. 7B) could be an intermediate[59], which intercepts the oxygen atom of nitroarene *N1* to afford the amino alcohol product **67**. In the presence of excess methanol as a competitive nucleophile, the model reaction exclusively yielded the amino alcohol product **67**, with no formation of the 1-methoxy-substituted alkyl aniline **67′**. This result suggested that the species *Int-1* is unlikely to be an intermediate in product formation. Furthermore, we considered the possibility that the *N*-(3-oxo-propyl) aniline species (*Int-2*, Fig. 7C) might serve as an intermediate[60], undergoing photocatalytic reduction to form the amino alcohol **112**. However, control experiment with 3-oxo-3-phenylpropyl aniline **111** under otherwise identical conditions did not produce the desired amino alcohol product, indicating that the species *Int-2* is unlikely to be an intermediate in the reaction.

(3) Probing the nitroarene-derived species for the reaction. Nitroarenes undergo photocatalytic reduction to generate various nitrogen-based species[61], which play a key role in the formation of *N*-arylamino alcohol products. The reaction of nitrobenzene (*N42*)

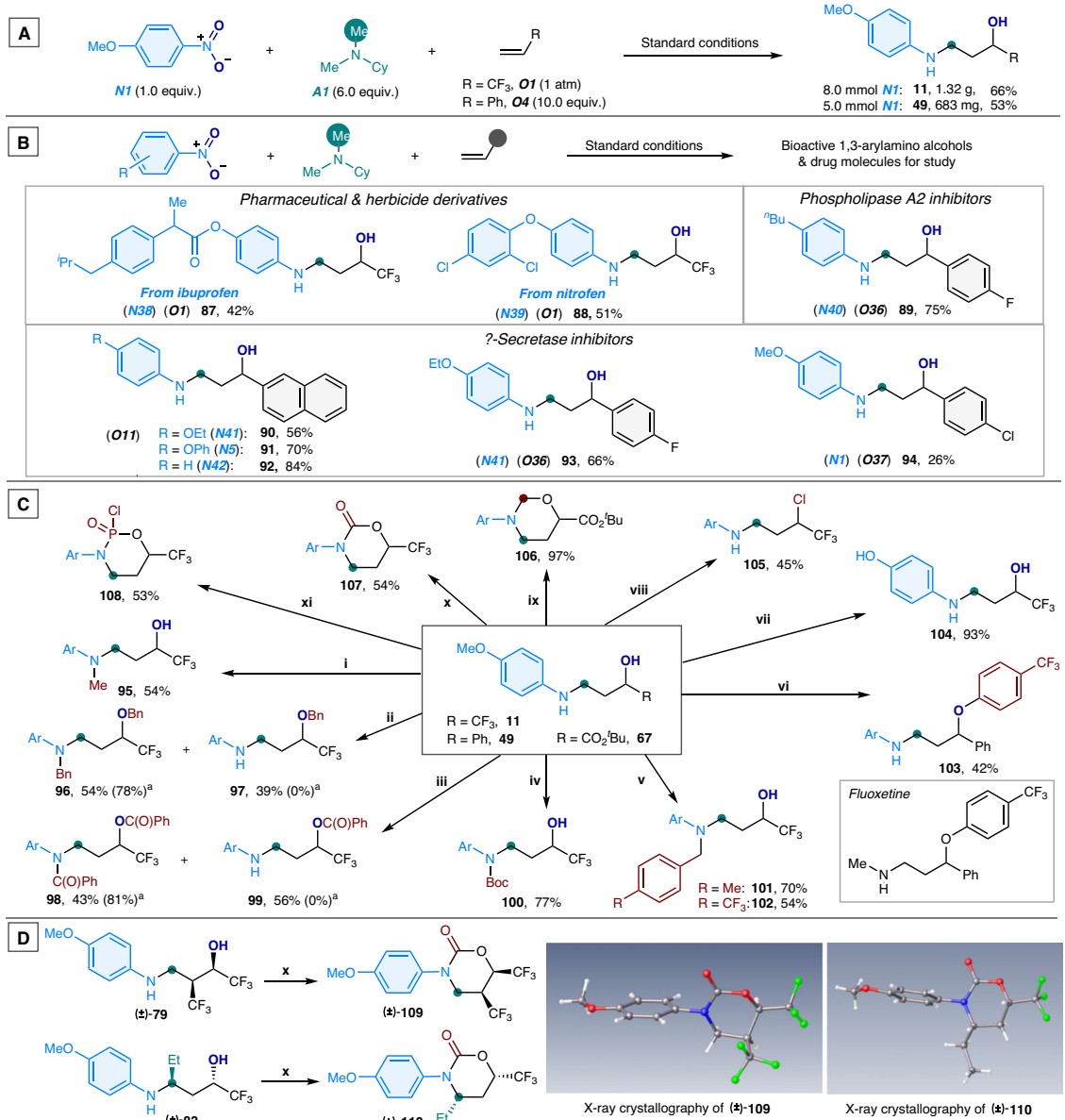

**Fig. 6 | Synthetic utility. A** Scalable synthesis of 3-arylamino alcohols. **B** Drug and argochemical-decorated γ-arylamino alcohols. **C** Derivatization of 3-arylamino alcohols. **D** Derivatization of 3-arylamino alcohol products (**79** and **80**) for stereochemical configuration. (**i**) MeI (1.2 equiv.), NaH (1.5 equiv.); (**ii**) BnBr (1.5 equiv.), NaH (1.5 equiv.); (**iii**) PhC(O)Cl (1.5 equiv.), Et₃N (2.0 equiv.); (**iv**) Boc₂O (1.5 equiv.); (**v**) *p*-tolualdehyde or 4-(trifluoromethyl)benzaldehyde (1.0 equiv.), NaBH(OAc)₃ (1.5 equiv.); (**vi**) 4-chlorobenzotrifluoride (1.1 equiv.), NaH (1.2 equiv.); (**vii**) BBr₃ (4.0 equiv.); (**viii**) PPh₃ (1.2 equiv.), TBAI (1,2 equiv.), 1,2-dichloroethane; (**ix**) HCHO (10 equiv.); (**x**) Triphosgene (1.0 equiv.), Et₃N (1.5 equiv.); (**xi**) POCl₃. ᵃ2.5 equiv. of electrophiles are used. X-ray crystallography data of **106** and **108** were obtained (see Supporting Information for details). Me methyl, Cy cyclohexyl, Ph phenyl, ʲPr isopropyl, ⁿBu n-butyl, Et ethyl, Boc *tert*-butoxycarbonyl, Ar aryl.

afforded the 3-phenylamino alcohol **113** in 94% yield (Fig. 7D, top right). During the reaction, nitrobenzene is sequentially reduced to nitrosobenzene (**N42-i**), *N*-phenyl hydroxylamine (**N42-ii**), azobenzene (**N42-iii**), azoxybenzene (**N42-iv**), *N,N'*-diphenyl hydrazine (**N42-v**), and aniline (**N42-vi**). Nitrobenzene may also react with a tertiary alkylamine (**A1**) under photoredox conditions to form *N*-phenyl imine (**N42-vii**) and *N*-methyl aniline (**N42-viii**), which could contribute to the reaction pathway. To examine the reactivity of these nitrogen-based species, they were subjected to the oxo-aminomethylation reaction under identical conditions (Fig. 7D, bottom). An exogenous 4-nitroanisole (**N1**) additive was introduced to act as the oxygen atom source for the hydroxyl group in the target amino alcohol product **113**, while also mimicking the redox conditions of the reaction. Only nitrosobenzene (**N42-i**) and *N*-phenyl hydroxylamine (**N42-ii**) reacted, affording **113** in 40% and 29% yields, respectively. The results suggested that

nitrosoarenes and *N*-aryl hydroxylamines are likely the key intermediates contributing to the formation of the amino alcohol products.

(4) Probing the reaction intermediates and co-products. To elucidate the mechanistic sequence of the oxo-aminomethylation reaction, we examined various intermediate species and co-products generated during the initial reaction stage using 4-nitroanisole (**N1**), *N,N*-dimethylcyclohexane (**A1**), and *tert*-butyl acrylate (**O21**) (Fig. 7E). Several species were detected using HRMS analysis, including nitrosoarene (**N1-i**), *N*-aryl *N*-hydroxyl aminal (**S1**), *N*-methylaminocyclohexane (**S2**), *N*-aryl nitrone (**S3**), and 2-aryl isoxazolidine (**S4**). We hypothesized that nitroarene is photocatalytically reduced to nitrosoarene (**N1-i**), followed by *N*-phenyl hydroxylamine (**N1-ii**)[61], both of which interact with tertiary alkylamine (**A1**) to form the aminal species **S1**[52]. **S1** undergoes deaminative C−N cleavage[51], yielding nitrone **S3** and secondary amine **S2**. **S3** then undergoes a facile cycloaddition with an

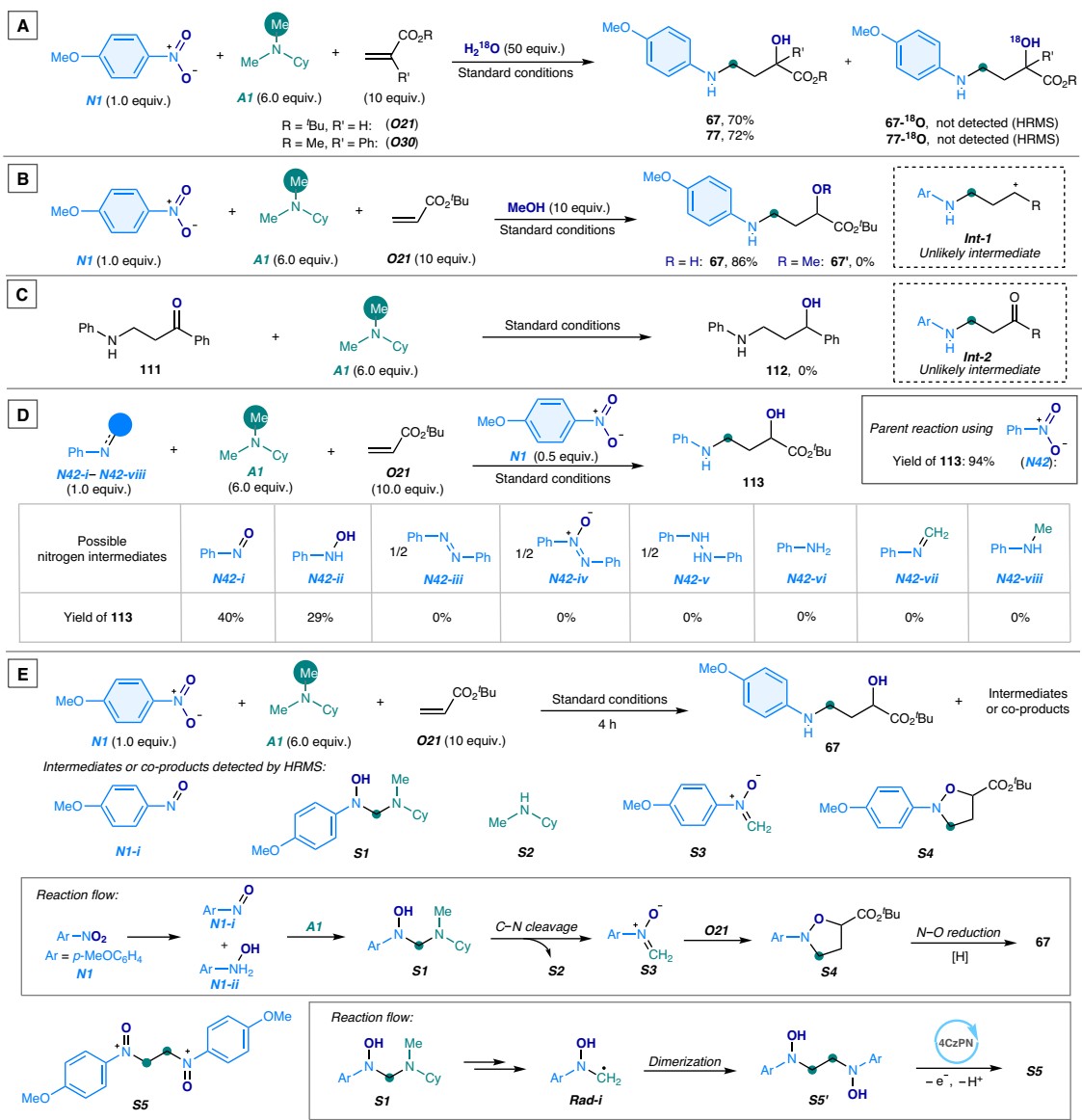

**Fig. 7 | Mechanistic study. A** Probing water as the hydroxyl source using [18]O-labeled water. **B** Probing γ-arylamino carbocation as a possible intermediate. **C** Probing *N*-(3-oxo-alkyl)aniline as a possible intermediate. **D** Probing nitrogen-based intermediates in the reaction. **E** Probing reaction intermediates at the initial stage of the reaction. Me methyl, Cy cyclohexyl, *t*Bu *tert*-butyl, Ph phenyl.

alkene to form isoxazolidine **S4**[49], which is subsequently reduced via N−O bond cleavage[49] to produce the amino alcohol **67**. Additionally, the oxo-substituted *N,N'*-diaryl ethylenediamine species (**S5**) was detected. This species likely arises from further transformation of aminal **S1**, which generates the *N*-aryl *N*-hydroxyl aminomethyl radical (**Rad-i**). **Rad-i** rapidly dimerizes to form the hydroxyl-substituted ethylenediamine species (**S5'**)[50], which, upon photocatalytic oxidation, regenerates **S5**. These detected species map the reaction pathway in the modular assembly of nitroarenes, tertiary alkylamines, and alkenes in the oxo-aminomethylation reaction.

(5) Probing the N−O bond cleavage step of isoxazolidine. The reductive N−O bond cleavage of isoxazolidines, as evidenced by the detected reaction intermediate **S4** (Fig. 7E; Figure S9, Supplementary Information), plays a key role in the formation of 3-arylamino alcohol products. To investigate this transformation, the reductive ring-opening of authentic isoxazolidine **114** was studied (Fig. 8). Under standard conditions, **114** underwent partial reduction, yielding 3-arylamino alcohol **63** in 48% yield, supporting its role as an intermediate (Fig. 8A (i)). When a protic source, trimethylamine

hydrochloride, was introduced, the reaction efficiency improved significantly, delivering **63** in 80% yield (Fig. 8A (ii)). This result highlighted the necessity of proton, generated during the photocatalytic oxidation of tertiary alkylamine **A1** and Hantzsch ester, in facilitating N−O bond cleavage. Next, we examined whether the ring-opening reaction of **114** is directly triggered by a photosensitizer (4CzPN) or a nickel species.

Under acidified conditions, the use of 4CzPN as the sole catalyst resulted in sluggish conversion, affording **63** in 27% yield (Fig. 8B (i)). In the absence of 4CzPN, the yield remained low (23%, Fig. 8B (ii)), indicating that a 4CzPN-mediated photocatalytic pathway is not operative. We then hypothesized that a nickel hydride species—either Ni[II](bipy)(H)(Cl) or Ni[I](bipy)H—might be responsible. These species could be formed via photoreduction of Ni(bipy)Cl₂ to Ni(0) or Ni(I), followed by interaction with protons or hydrogen atoms derived from tertiary alkylamine **A1** or Hantzsch ester. To test this, we conducted the reaction in the presence of in-situ generated Ni−H species using Ni(bipy)Cl₂ and sodium borohydride[62,63]. Without blue light irradiation, isoxazolidine **114** converted efficiently to **63** in 72% yield (Fig. 8C (i)).

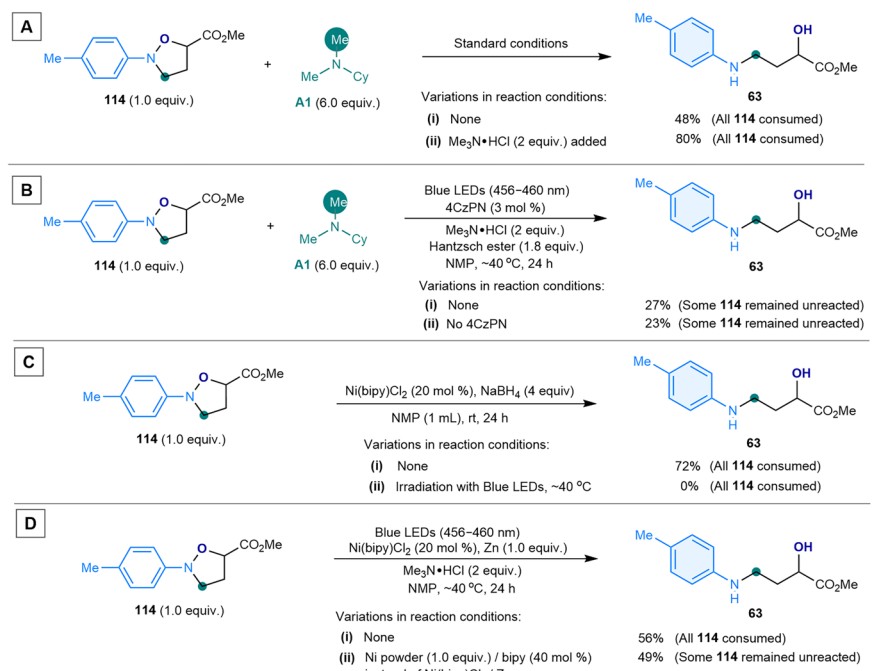

**Fig. 8 | Probing the reductive N–O bond cleavage of isoxazolidine. A** Effect of protonation on the N–O bond cleavage of isoxazolidine. **B** Reactivity of the photocatalyst toward N–O bond cleavage of isoxazolidine. **C** Reactivity of the Ni–H species toward N–O bond cleavage of isoxazolidine. **D** Reactivity of the Ni⁰ species toward N–O bond cleavage of isoxazolidine. Me methyl, Cy cyclohexyl.

However, under blue LED irradiation, **114** decomposed without generating the amino alcohol product (Fig. 8C (ii)), suggesting that Ni–H species is not the primary reductant responsible for N–O bond cleavage under the light-driven conditions. Finally, we investigated the role of Ni(0) species, Ni⁰(bipy), as a potential mediator for ring opening. Using a catalytic amount of Ni(bipy)Cl₂ in combination with zinc powder as the terminal reductant significantly enhanced N–O bond cleavage, affording the amino alcohol product in 56% yield (Fig. 8D (i)). Similarly, employing stoichiometric nickel powder along with a substoichiometric amount of 2,2′-bipyridyl ligand (40 mol %) resulted in a comparable yield (49%, Fig. 8D (ii)). Control experiments confirmed that Ni⁰(bipy) is likely the key reductant[64,65], facilitating the conversion of isoxazolidines into amino alcohols.

(6) Stern-Volmer Quenching Study. The photocatalyst 4CzPN undergoes photoexcitation to its triplet state (4CzPN*). This long-lived, high-energy species can be either oxidatively or reductively quenched by various reactants and reagents, including nitroarene, tertiary alkylamine, alkene, Hantzsch ester, and Ni(bipy)Cl₂. In the Stern-Volmer quenching study (Figures S11–S16, Supplementary Information), all the quenching species − 4-nitroanisole (**N1**), *N,N*-dimethylaminocyclohexane (**A1**), *tert*-butyl acrylate (**O21**), Hantzsch ester (**HE**), and Ni(bipy)Cl₂ − were able to quench the photoexcited photocatalyst, with Hantzsch ester being the most effective quenching agent. The results suggested that 4CzPN* is likely reduced by Hantzsch ester to form a highly reducing photocatalyst radical anion species (4CzPN·⁻), which plays a key role in initiating the oxy-aminomethylation reactions and triggering subsequent redox processes that ultimately lead to the formation of the 3-amino alcohol products.

Based on the mechanistic experimental results, we propose a viable mechanism for the metallaphotocatalytic oxo-aminomethylation reaction (Fig. 9A). Upon blue light irradiation, the photosensitizer 4CzPN is excited to its high-energy, redox-active state (4CzPN*) {$E_{1/2}^{red}$ [4CzPN*/ 4CzPN·⁻] = +1.40 vs SCE}[66]. This species is readily reduced by Hantzsch ester (**HE**) to form the organo radical anion species 4CzPN·⁻, while **HE** is oxidized to its cationic form (**HE·⁺**) {$E_{1/2}^{red}$ [**HE·⁺**/**HE**] =

+0.89 V vs SCE}[67]. Additionally, *N,N*-dimethylaminocyclohexane (**A1**) is oxidized by 4CzPN* to generate the amino radical cation (**A1′**) {$E_{1/2}^{red}$ [ⁱPrMe₂N·⁺/ⁱPrMe₂N] = +0.72 V vs SCE}[68], which rapidly undergoes deprotonation to yield the *N*-cyclohexyl-*N*-methyl-aminomethyl radical (**Rad-ii**). The 4CzPN·⁻ species is highly reducing {$E_{1/2}^{red}$ [4CzPN·⁻/ 4CzPN] = −1.16 vs SCE}[66], facilitating the reduction of Niⁱⁱ(bipy)Cl₂ to nickel(I) and nickel(0) species, Niⁱ(bipy)Cl (**Niⁱ**) and Ni⁰(bipy) (**Ni⁰**), respectively {$E_{1/2}^{red}$ [Niⁱⁱ(bipy)Cl₂/Niⁱ(bipy)Cl] ~ −1.12 vs SCE}[69]; {$E_{1/2}^{red}$ [Niⁱ(bipy)Cl/Ni⁰(bipy)] ~ −0.6 vs SCE}[70]. Meanwhile, nitroarene (**N1**) is reduced by 4CzPN·⁻ to its radical anion (**N1·⁻**)[71] {$E_{1/2}^{red}$ [*p*-MeOC₆H₄NO₂/ *p*-MeOC₆H₄NO₂·⁻] ~ −0.88 vs SCE}[52], which undergoes further photocatalytic reduction and water elimination to form nitrosoarene (**N1-i**)[61]. The electrophilic nitrosoarene[72] then interacts with the nucleophilic aminomethyl radical (**Rad-ii**)[72] to form an aminal-based oxygen radical (**Rad-iii**), which is subsequently reduced to yield *N*-aryl-*N*-hydroxyl aminal (**S1**)[51,52]. As an alternative pathway, nitrosoarene can be further reduced to *N*-aryl hydroxylamine (**N1-ii**)[61], which interacts with **Niⁱⁱ** and the aminomethyl radical (**Rad-ii**) to form the alkyl- and amino-coordinated nickel(III) complex intermediate (**S6**), followed by reductive elimination to yield aminal **S1**.

*N*-aryl-*N*-hydroxyl aminal (**S1**) undergoes facile proton-catalyzed elimination of the secondary amine **S2** to afford nitrone **S3**. Nitrone **S3** undergoes cycloaddition with an alkene (e.g., **O21**), yielding *N*-aryl isoxazolidine (**S4**)[49]. Protonation at the more basic nitrogen of the isoxazolidine, followed by coordination with a nickel(0) complex, leads to the formation of species **S4′**. The inner-sphere electron transfer from nickel(0) to the electron-deficient, protonated isoxazolidine is highly favorable, facilitating N–O bond cleavage[64,65] and leading to the formation of nickel−(arylamino)alkoxide (**67′**). This intermediate then undergoes protonation and demetalation, affording the final 3-arylamino alcohol product (**67**).

We propose that the concerted cycloaddition of *N*-aryl nitrones with alkenes constitutes the predominant reaction pathway, leading to the stereospecific formation of isoxazolidines and, subsequently, 3-arylamino alcohols. This simultaneous transformation plays a critical role in reactions involving internal alkenes (**O32**, Fig. 4) and tertiary

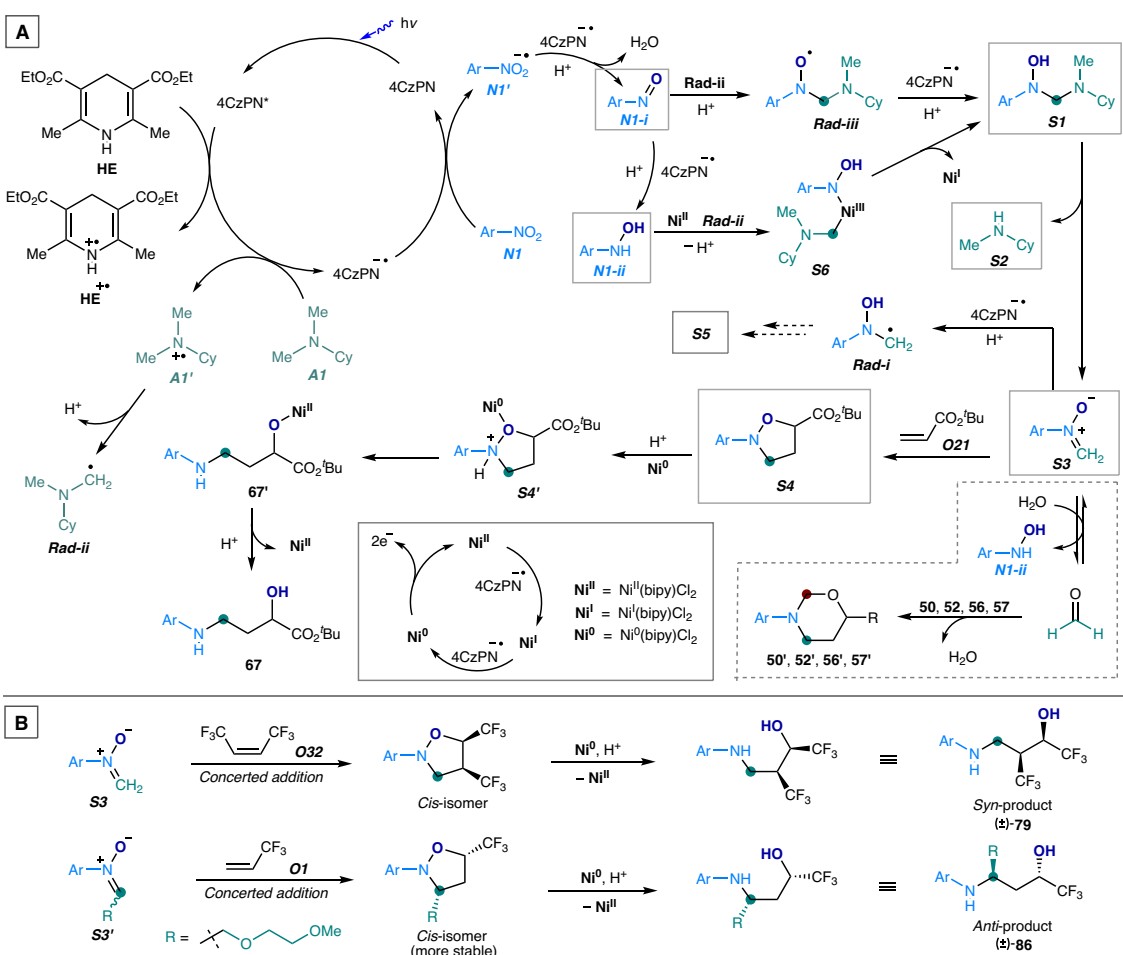

**Fig. 9 | Proposed mechanism. A** Proposed mechanism of the metallaphotocatalytic oxo-aminomethylation of alkenes. **B** Proposed mechanism for the stereospecific formation of 3-arylamino propanol products via a concerted cycloaddition pathway. Et ethyl, Me methyl, Cy cyclohexyl, *t*Bu *tert*-butyl, Ar = 4-methoxyphenyl.

alkylamines (**A5**–**A8**, Fig. 5), consistently affording single diastereomers of amino alcohols (**79**, **83**–**86**) bearing two stereogenic centers. For example, nitrone **S3** undergoes a concerted cycloaddition with a di-trifluoromethyl-substituted alkene **O32** to form an isoxazolidine bearing two *cis*-oriented vicinal CF₃ groups (Fig. 9B, top), thus yielding the *syn*-disubstituted amino alcohol product **79**. Similarly, the nitrone **S3′**, derived from tertiary alkylamine **A8**, reacts with 3,3,3-trifluoropropene **O1** via a concerted cycloaddition to generate a thermodynamically favored isoxazolidine in which the bulky 3-aliphatic (R) and 5-CF₃ groups occupy equatorial and *cis*-positions (Fig. 9B, bottom), thereby affording the *anti*-disubstituted amino alcohol product **86**. The alternative formation of isoxazolidine intermediate via a stepwise radical addition–cyclization mechanism, involving an *N*-aryl-*N*-hydroxy aminomethyl radical (**Rad-i**) properly generated through photocatalytic, proton-promoted reduction of nitrone **S3**, appears unlikely, since such a pathway would afford a mixture of diastereomeric products. In reactions with certain alkenes (**O5**, **O7**, **O11** and **O12**, Fig. 4), the *N*-aryl nitrone intermediates undergo hydrolysis to liberate formaldehyde and *N*-aryl hydroxylamines (Fig. 9A, bottom right). The subsequent condensation of formaldehyde with the 3-arylamino alcohol products (**50**, **52**, **56** and **57**) leads to the formation of 1,3-oxazinanes (**50′**, **52′**, **56′**, **57′**) as minor co-products.

In summary, we have successfully developed a metallaphotoredox-catalyzed multicomponent oxo-aminomethylation reaction that leverages nitroarenes as dual nitrogen and oxygen sources in combination with tertiary alkylamines and alkenes. This modular and efficient synthetic strategy features a broad substrate scope and excellent functional group tolerance, enabling the streamlined synthesis of structurally diverse 3-arylamino alcohols. Furthermore, the versatility of these amino alcohols allows for diverse post-synthetic modifications, facilitating the creation of advanced derivatives with expanded functional applications. We anticipate that this general oxo-aminomethylation protocol will inspire further exploration of nitroarene-based transformations and contribute to the discovery of novel bioactive compounds with enhanced structural complexity and therapeutic potential.

## Methods

### Oxy-aminomethylation reaction with gaseous alkenes

An oven-dried, transparent 20 mL Teflon screw-capped Schlenk tube equipped with a stir bar was sequentially charged with nitroarene (1.0 equiv., 0.10 mmol), 4CzPN (3 mol%, 0.0030 mmol), Ni(bipy)Cl₂ (20 mol%, 0.020 mmol), and Hantzsch ester (**HE**, 1.8 equiv., 0.18 mmol). Dried *N*-methyl-2-pyrrolidone (NMP, 1.0 mL) was then transferred into the tube via syringe. Subsequently, *N*,*N*-dimethylcy-clohexylamine (6.0 equiv., 0.60 mmol) was transferred into the tube via syringe. The resulting mixture was degassed via blowing with a balloon filled with 3,3,3-trifluoropropene (~1 L gas) for 2 min, after which time the tube was quickly capped with a Teflon screw cap such that it was filled with 3,3,3-trifluoropropene in atmospheric pressure. The reaction mixture was vigorously stirred and irradiated using 30 W blue LEDs (λ = 455 – 460 nm) for 24 h, during which time the proximal temperature was controlled at approximately 40 °C via cooling with

fans. At this point, the reaction mixture was diluted with ethyl acetate (100 mL) and washed with water (50 mL × 2). The organic fraction was further dried with anhydrous $Na_2SO_4$ and concentrated *in vacuo* with the aid of rotary evaporator. The residue was purified by preparative thin-layer chromatography using a mixture of petroleum ether and ethyl acetate as an eluent to afford the 3-arylamino alcohol product.

## Oxy-aminomethylation reaction with non-gaseous alkenes

An oven-dried, transparent 20 mL Teflon screw-capped Schlenk tube equipped with a stir bar was sequentially charged with nitroarene (1.0 equiv., 0.10 mmol), 4CzPN (3 mol %, 0.0030 mmol), Ni(bipy)$Cl_2$ (20 mol %, 0.020 mmol), and Hantzsch ester (**HE**, 1.8 equiv., 0.18 mmol). The reaction mixture was degassed and backfilled with argon three times. Under a positive argon pressure, dried *N*-methyl-2-pyrrolidone (NMP, 1.0 mL), alkene (10.0 equiv., 1.0 mmol), and *N,N*-dimethylcyclohexylamine (6.0 equiv., 0.60 mmol) were added via syringe. The reaction mixture was vigorously stirred and irradiated using 30 W blue LEDs ($\lambda$ = 455 – 460 nm) for 24 h, during which time the proximal temperature was controlled at approximately 40 °C via cooling with fans. At this point, the reaction mixture was diluted with ethyl acetate (100 mL) and washed with water (50 mL × 2). The organic fraction was further dried with anhydrous $Na_2SO_4$ and concentrated *in vacuo* with the aid of rotary evaporator. The residue was purified by preparative thin-layer chromatography using a mixture of petroleum ether and ethyl acetate as an eluent to afford the 3-arylamino alcohol product.

## Data availability

Detailed experimental procedures, analytical methods, and complete spectral data are provided in the Supplementary Information. The crystallographic data generated in this study have been deposited in the Cambridge Crystallographic Data Center (CCDC) under deposition numbers CCDC 2426457 (**26**), CCDC 2426464 (**49**), CCDC 2426459 (**71**), CCDC 2464714 (**86**), CCDC 2426465 (**106**), CCDC 2426466 (**108**), CCDC 2464379 (**109**), and CCDC 2464376 (**110**), and are available free of charge at https://www.ccdc.cam.ac.uk/structures/. Data supporting the findings of this manuscript are also available from the corresponding author upon request.

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

## Acknowledgements

We thank the National Natural Science Foundation of China [Nos. 92156025 (J.-A.M.), 22361142832 (J.-A.M.), 21971186 (C.W.C.), and 22271216 (C.W.C.)] for financial support.

## Author contributions

T.Z. discovered the reactions, optimized the reactions and studied the reaction scope, synthetic utility and reaction mechanisms. C.W.C. wrote the manuscript with help and suggestions from T.Z., J.N. and J.-A.M. C.W.C. and J.-A.M. conceived the project, directed the research and designed the experiments.

## Competing interests

The authors declare no competing interests.
