## [Transparent Peer Review file · Nature Communications]

Harnessing nitroarenes as nitrogen and oxygen sources for general oxo-aminomethylation of alkenes

Corresponding Author: Professor Chi Wai Cheung

Version 0:

Reviewer comments:

Reviewer #1

(Remarks to the Author)

This manuscript by Ma, Cheung and co-workers describes a novel and practical method for the synthesis of 3-amino alcohol derivatives via metallaphotoredox-catalyzed multicomponent oxo-aminomethylation of alkenes with nitroarenes and tertiary alkylamines. The readily accessible nitroarenes, in this coupling protocol, are used as both nitrogen and oxygen sources, constructing structurally diverse 3-amino alcohols that are prevalent in bioactive molecules. As shown in Figures 2 and 3, this multi-component transformations involving an open-shell pathway feature broad substrate scope and functional group tolerance. The yields provided in this radical protocol are satisfactory with respect to the components of both nitroarenes and alkenes, whereas the scope of tertiary alkylamines is still limited. The expedient manipulations for gram-scale synthesis and follow-up chemistry further demonstrate the practical relevance of this radical cascade protocol. The outcomes of mechanistic studies give a comprehensive understanding of the reaction details, and the proposed mechanism is reasonable, according to both experimental evidence and their previous reports. The supporting information is well prepared, all of unknown compounds are characterized properly. Therefore, this manuscript is recommended for publication in Nature Communications after addressing the following issues:

1. In reaction optimization, "4CzPIN" should be "4CzIPN"
2. In Figure 6, there is no [Ir] photocatalyst in the process of transforming S5' into S5.
3. In line 410, the "aminomethyl radical" is (Rad-ii), not (A1').
4. In Figure 8, nickel cycle should be reduced by 4CzPN^{-•}.
5. Please describe in the manuscript the details for the generation of 1,3-oxazinanes as co-products for some substrates (O5, O7, O11 and O12, Figure 3).
6. The authors should discuss the results of electron-rich alkenes in this transformation.

Reviewer #2

(Remarks to the Author)

The authors previously reported related transformation, four-component coupling of nitroarenes, alkenes, amines, and redox active esters, in Nature Commun 2024, 15, 9926 (ref 52). In this manuscript, the authors performed three-component coupling reaction in the absence of redox active esters, leading to different products. The broad substrate scope of nitroarenes and alkenes has been achieved, and functionalized 3-arylamino alcohols were obtained efficiently. Furthermore, amine scope was expanded as shown in Figure 4, providing products in diastereoselective manner. The basic reaction design is similar to their previous report in ref 52, and so the conceptual novelty may be somewhat limited (compared with their previous reports). On the other hand, the synthetic utility is extremely high, and so the reviewer think the manuscript is still worthy for publication in Nature Commun. But, major revisions in responses to the requests listed below should be addressed.

Revision requests:

- 1) The phrase "both the nickel and Hantzsch ester were essential for optimal performance" in page 5, should be modified. Based on the result in Table 1, entry 14: Hantzsch ester only slightly improved the catalytic efficiency. I am afraid the difference is too small to say it is essential.
- 2) Table 1, entry 14: The catalyst is shown as NiCl₂Bipy in entry 14, while in other entries shown as Ni(Bipy)Cl₂. What is the

difference? If the difference is in situ mixed metal source and ligand (entry 14) and pre-formed Ni(II)-Bipy complex (other entries), then the control experiment in entry 14 should be re-performed under identical conditions using the same catalyst.

3) Figure 3, The scope of alkenes: The results using alkyl substitute alkenes should be included in Figure 3.

4) Figure 4: Dr is not shown in Figure 4. If the authors obtained products as single isomer as described so in the conclusion, note on diastereoselectivity should be added in Figure 4, like dr > 20:1.

5) Stereochemistry of products in Figure 4 should be clearly determined. Syn or anti? The stereochemistry of products would be useful for mechanistic consideration.

6) Figure 8, proposed mechanism:

The authors performed several mechanistic studies, similar to ref 52, and some plausible intermediates were detected by MS. That is nice.

On the other hand, some of the proposed role of Ni catalyst is too much speculative. For example, from intermediate S1 to S3, the Ni-catalyzed process may not be required. Aminal S1 can be simply transformed into S3 and S2 in the presence of some proton source. So, I think the intermediate S7 (path from S1 to S7 and S3) should be deleted.

Reviewer #3

(Remarks to the Author)

In this manuscript, Ma and co-authors present a metallaphotoredox-catalyzed multicomponent oxo-aminomethylation approach employing nitroarenes, tertiary alkylamines, and alkenes for the synthesis of 3-arylamino alcohols. Notably, nitroarenes are utilized as bifunctional nitrogen and oxygen sources, enabling a cascade transformation that is both mechanistically intricate and synthetically valuable. The protocol demonstrates broad substrate scope, high functional group tolerance, and excellent regioselectivity, affording access to a diverse array of secondary and tertiary amino alcohols. Furthermore, the use of trialkylamines facilitates the incorporation of a wide range of alkyl substituents.

The synthetic applicability of the methodology is convincingly illustrated through gram-scale reactions and downstream modifications to form pharmaceutically relevant compounds. The manuscript also includes a set of mechanistic studies aimed at elucidating the reaction pathway. Overall, the work is of high scientific merit and warrants publication in Nature Communications after addressing the following issues:

Points for Revision

* The clarity of several figures could be improved. For instance, in Figure 2, the numerical labeling of all nitroarene substrates (N) and corresponding products results in visual complexity that detracts from interpretability. Similar issues are observed in other figures.

* Figure 1 is overly dense and may benefit from simplification or division into subpanels to improve conceptual clarity and readability.

* The use of 20 mol% nickel catalyst raises some concern, particularly in examples where yields fall below 50%, approaching stoichiometric levels. Have the authors attempted catalyst loading optimization? A brief discussion of such efforts and any limitations encountered would strengthen the manuscript.

* The mechanistic proposal depicted in Figure 8, while comprehensive, is highly complex. In the absence of supporting DFT calculations or further experimental validation, it may be advisable to streamline the mechanistic discussion or relocate the figure to the Supporting Information.

* The name of the photocatalyst "4CzIPN" is consistently misspelled throughout the manuscript and in the figures. This should be corrected for consistency and accuracy.

Version 1:

Reviewer comments:

Reviewer #1

(Remarks to the Author)

The authors provided an improved manuscript that has comprehensively addressed the issues raised during the peer-review process. Therefore, I strongly recommend this manuscript for publication in its current form.

Reviewer #2

(Remarks to the Author)

Revisions in responses to the requests in previous round of peer-review are appropriate. I recommend publication of this manuscript without further revisions.

Reviewer #3

(Remarks to the Author)

In this revised manuscript, the authors have appropriately addressed the concerns raised by me and other reviewers. I recommend the manuscript for publication in Nature Communications.

Typos:

Figure 9: the subtitle in both the text and figure should read “metallaphotocatalytic” instead of “metallophotocatalytic.”

Response to the Referees

We sincerely thank the Reviewers for their time and effort in evaluating our manuscript. We have carefully revised the paper in accordance with their insightful and constructive suggestions. Below is a point-by-point response to each of the comments raised.

Referee: 1

Recommendation: Recommended for publication in *Nature Communications* after revision.

Comment 1: This manuscript by Ma, Cheung and co-workers describes a novel and practical method for the synthesis of 3-amino alcohol derivatives via metallaphotoredox-catalyzed multicomponent oxo-aminomethylation of alkenes with nitroarenes and tertiary alkylamines. The readily accessible nitroarenes, in this coupling protocol, are used as both nitrogen and oxygen sources, constructing structurally diverse 3-amino alcohols that are prevalent in bioactive molecules. As shown in Figures 2 and 3, this multi-component transformations involving an open-shell pathway feature broad substrate scope and functional group tolerance. The yields provided in this radical protocol are satisfactory with respect to the components of both nitroarenes and alkenes, whereas the scope of tertiary alkylamines is still limited. The expedient manipulations for gram-scale synthesis and follow-up chemistry further demonstrate the practical relevance of this radical cascade protocol. The outcomes of mechanistic studies give a comprehensive understanding of the reaction details, and the proposed mechanism is reasonable, according to both experimental evidence and their previous reports. The supporting information is well prepared, all of unknown compounds are characterized properly. Therefore, this manuscript is recommended for publication in *Nature Communications* after addressing the following issues:

Our response: We sincerely appreciate the Reviewer's encouraging comments and positive evaluation of our work.

Comment 2: 1. In reaction optimization, "4CzPIN" should be "4CzIPN"

Our response: We have corrected the name of the photocatalyst throughout the revised manuscript and figures to the correct term, 4CzPN (PC4).

Comment 3: 2. In Figure 6, there is no [Ir] photocatalyst in the process of transforming S5' into S5.

Our response: We have revised Figure 7 in the updated manuscript to accurately indicate that the

transformation from **S5'** to **S5** involves 4CzPN (**PC4**) rather than the Ir photocatalyst.

Comment 4: 3. In line 410, the “aminomethyl radical” is (Rad-ii), not (A1').

Our response: We have corrected the text in the revised manuscript to refer to the aminomethyl radical as **Rad-ii** instead of **A1'**.

Comment 5: 4. In Figure 8, nickel cycle should be reduced by 4CzPN•.

Our response: We have revised the figure (currently Figure 9A in the revised manuscript) to explicitly indicate that the nickel catalyst is reduced by 4CzPN•.

Comment 6: 5. Please describe in the manuscript the details for the generation of 1,3-oxazinanes as co-products for some substrates (**O5**, **O7**, **O11** and **O12**, Figure 3).

Our response: We have added a discussion to the revised manuscript, explaining that the formation of 1,3-oxazinanes arises from the condensation of formaldehyde, generated *in situ* via the hydrolysis of *N*-aryl nitron intermediate, with the 3-arylamino alcohol products derived from these specific alkenes.

We have added the following descriptions to the revised manuscript:

“Reactions with certain alkenes (**O5**, **O7**, **O11** and **O12**) yield the corresponding 1,3-oxazinanes in yields ranging from 23% to 45%, likely due to the cyclization of the 1,3-arylamino alcohol products with formaldehyde generated *in situ* (vide infra).”

“In reactions with certain alkenes (**O5**, **O7**, **O11** and **O12**, Figure 4), the *N*-aryl nitron intermediate undergoes hydrolysis to liberate formaldehyde and *N*-aryl hydroxylamine (Figure 9A, bottom right). The subsequent condensation of formaldehyde with the 3-arylamino alcohol products (**50**, **52**, **56** and **57**) leads to the formation of 1,3-oxazinanes (**50'**, **52'**, **56'** and **57'**) as minor co-products.”

Comment 7: 6. The authors should discuss the results of electron-rich alkenes in this transformation.

Our response: We have added the following description to the revised manuscript, noting that electron-rich alkenes (e.g., 4-phenylbut-1-ene, vinyl acetate, ethoxyethene) did not yield the desired 3-arylamino alcohols:

“Furthermore, electron-rich alkenes, such as 4-phenylbut-1-ene, vinyl acetate, and ethoxyethene, failed to undergo the reaction to afford the corresponding 1,3-arylamino alcohols.”

Referee: 2

Recommendation: Recommended for publication in *Nature Communications* after major revision.

Comment 1: The authors previously reported related transformation, four-component coupling of nitroarenes, alkenes, amines, and redox active esters, in *Nature Commun* 2024, 15, 9926 (ref 52). In this manuscript, the authors performed three-component coupling reaction in the absence of redox active esters, leading to different products. The broad substrate scope of nitroarenes and alkenes has been achieved, and functionalized 3-arylamino alcohols were obtained efficiently. Furthermore, amine scope was expanded as shown in Figure 4, providing products in diastereoselective manner. The basic reaction design is similar to their previous report in ref 52, and so the conceptual novelty may be somewhat limited (compared with their previous reports). On the other hand, the synthetic utility is extremely high, and so the reviewer think the manuscript is still worthy for publication in *Nature Commun*. But, major revisions in responses to the requests listed below should be addressed.

Our response: We sincerely appreciate the Reviewer's comments and positive evaluation of our work.

Comment 2: Revision requests: 1) The phrase "both the nickel and Hantzsch ester were essential for optimal performance" in page 5, should be modified. Based on the result in Table 1, entry 14: Hantzsch ester only slightly improved the catalytic efficiency. I am afraid the difference is too small to say it is essential.

Our response: We have revised the manuscript to clarify that both nickel and Hantzsch ester are required for optimal efficiency, but only the nickel catalyst is strictly essential.

We have added the following description to the revised manuscript:

"Control experiments confirmed that the nickel complex was essential for optimal reaction performance (Entry 15), while the addition of Hantzsch ester could further improve the product yield (Entry 16)."

Comment 3: 2) Table 1, entry 14: The catalyst is shown as NiCl₂Bipy in entry 14, while in other entries shown as Ni(Bipy)Cl₂. What is the difference? If the difference is in situ mixed metal source and ligand (entry 14) and pre-formed Ni(II)-Bipy complex (other entries), then the control experiment in entry 14 should be re-performed under identical conditions using the same catalyst.

Our response: We apologize for the typo. The correct notation is Ni(Bipy)Cl₂ complex rather than NiCl₂bipy. This has been corrected in Table 1, Entry 16 of the revised manuscript.

Comment 4: 3) Figure 3, The scope of alkenes: The results using alkyl substitute alkenes should be included in Figure 3.

Our response: We have added a sentence in the revised manuscript, noting that electron-rich alkenes (e.g., 4-phenylbut-1-ene, vinyl acetate, ethoxyethene) did not yield the desired 3-arylamino alcohols.

We have added the following description to the revised manuscript:

“Furthermore, electron-rich alkenes, such as 4-phenylbut-1-ene, vinyl acetate, and ethoxyethene, failed to undergo the reaction to afford the corresponding 1,3-arylamino alcohols.”

Comment 5: 4) Figure 4: Dr is not shown in Figure 4. If the authors obtained products as single isomer as described so in the conclusion, note on diastereoselectivity should be added in Figure 4, like $dr > 20:1$.

Our response: Based on ^1H , ^{13}C , and ^{19}F NMR spectroscopy, only one set of signals was observed for the 3-arylamino alcohol products bearing two stereocenters (compounds **79**, **83–86**), suggesting the formation of single diastereomers. We have added the diastereomeric ratios ($dr > 20:1$) for these compounds to Figures 4 and 5 in the revised manuscript.

Comment 6: 5) Stereochemistry of products in Figure 4 should be clearly determined. Syn or anti? The stereochemistry of products would be useful for mechanistic consideration.

Our response: Based on X-ray crystallographic analysis, the stereochemistry of compound **86** (Figure 5, revised manuscript) was unambiguously determined to be an *anti*-substituted structure, suggesting that compounds **83–86** share the same *anti*-configuration. Furthermore, the X-ray crystal structure of the annulation product of compound **83** (compound **110**, Figure 6) confirmed this *anti*-configuration. In contrast, the X-ray structure of the annulation product of compound **79** (compound **109**, Figure 6) indicated that compound **79** adopts a *syn*-configuration. Stereochemical structures have been provided in the revised manuscript.

Comment 7: 6) Figure 8, proposed mechanism: The authors performed several mechanistic studies, similar to ref 52, and some plausible intermediates were detected by MS. That is nice. On the other hand, some of the proposed role of Ni catalyst is too much speculative. For example, from intermediate S1 to S3, the Ni-catalyzed process may not be required. Aminal S1 can be simply transformed into S3 and S2 in the presence of some proton source. So, I think the intermediate S7 (path from S1 to S7 and S3) should be deleted.

Our response: We thank the Reviewers for this suggestion. We have removed the speculative reaction mechanism involving intermediate S7. Nitron S3 is now described as being generated via proton-catalyzed elimination of the secondary amine S2 from aminal S1.

We have added the following description to the revised manuscript:

“*N*-aryl-*N*-hydroxyl aminal (**S1**) undergoes facile proton-catalyzed elimination of the secondary amine **S2** to afford nitron **S3**.”

Referee: 3

Recommendation: Recommended for publication in *Nature Communications* after revision.

Comment 1: In this manuscript, Ma and co-authors present a metallaphotoredox-catalyzed multicomponent oxo-aminomethylation approach employing nitroarenes, tertiary alkylamines, and alkenes for the synthesis of 3-arylamino alcohols. Notably, nitroarenes are utilized as bifunctional nitrogen and oxygen sources, enabling a cascade transformation that is both mechanistically intricate and synthetically valuable. The protocol demonstrates broad substrate scope, high functional group tolerance, and excellent regioselectivity, affording access to a diverse array of secondary and tertiary amino alcohols. Furthermore, the use of trialkylamines facilitates the incorporation of a wide range of alkyl substituents. The synthetic applicability of the methodology is convincingly illustrated through gram-scale reactions and downstream modifications to form pharmaceutically relevant compounds. The manuscript also includes a set of mechanistic studies aimed at elucidating the reaction pathway. Overall, the work is of high scientific merit and warrants publication in *Nature Communications* after addressing the following issues:

Our response: We thank the Reviewer for their positive assessment of our study's scope, mechanistic detail, and practical utility.

Comment 2: Points for Revision* The clarity of several figures could be improved. For instance, in Figure 2, the numerical labeling of all nitroarene substrates (N) and corresponding products results in visual complexity that detracts from interpretability. Similar issues are observed in other figures.

Our response: We have italicized the numerical labels of reaction substrates (*N*), alkenes (*O*), tertiary alkylamines (*A*), and intermediates (*Int*) in all figures and throughout the revised manuscript and Supplementary Information to facilitate identification.

Comment 3: * Figure 1 is overly dense and may benefit from simplification or division into subpanels to improve conceptual clarity and readability.

Our response: We have split Figure 1 into two clearer panels (now Figures 1 and 2 in the revised manuscript) to enhance conceptual clarity and reduce visual crowding.

Comment 4: * The use of 20 mol% nickel catalyst raises some concern, particularly in examples where yields fall below 50%, approaching stoichiometric levels. Have the authors attempted catalyst loading optimization? A brief discussion of such efforts and any limitations encountered would strengthen the manuscript.

Our response: We attempted to reduce the catalyst loadings to 10 mol% and 15 mol % during the

reaction optimization (Table 1), but this resulted in diminished yields (63% and 62%), lower than the yields using 20 mol % (71%). Therefore, 20 mol % of Ni salt was employed to carry out the reaction scope study.

We have added the following description to the revised manuscript:

“Since the use of lower loadings of the Ni catalyst led to diminished product yields (Entries 11 and 12), 20 mol % of the Ni catalyst was employed for further studies.”

Comment 5: * The mechanistic proposal depicted in Figure 8, while comprehensive, is highly complex. In the absence of supporting DFT calculations or further experimental validation, it may be advisable to streamline the mechanistic discussion or relocate the figure to the Supporting Information.

Our response: We appreciate the reviewer’s suggestion to streamline the mechanistic discussion. Based on Reviewer 2’s comments, we have simplified the mechanistic proposal in the revised manuscript (now presented as Figure S9A) to enhance clarity. We believe that retaining this figure in the main text will facilitate readers’ understanding of the reaction mechanism, and therefore we prefer to keep it in the main text rather than moving it to the Supplementary Information.

Comment 6: * The name of the photocatalyst “4CzIPN” is consistently misspelled throughout the manuscript and in the figures. This should be corrected for consistency and accuracy.

Our response: We thank the Reviewers for pointing out this error. We have corrected all instances of the photocatalyst name throughout the revised manuscript and figures to 4CzPN.

Response to the Referees

We sincerely thank the Reviewers for their time and effort in evaluating our manuscript. We have carefully revised the paper in accordance with their insightful and constructive suggestions. Below is a point-by-point response to each of the comments raised.

Referee: 1

Recommendation: Recommendation for publication.

Comment 1: The authors provided an improved manuscript that has comprehensively addressed the issues raised during the peer-review process. Therefore, I strongly recommend this manuscript for publication in its current form.

Our response: We sincerely thank the Reviewer for the positive assessment and recommendation for publication.

Referee: 2

Recommendation: Recommendation for publication.

Comment 1: Revisions in responses to the requests in previous round of peer-review are appropriate. I recommend publication of this manuscript without further revisions.

Our response: We are grateful for the Reviewer's supportive evaluation and recommendation for publication.

Referee: 3

Recommendation: Recommendation for publication.

Comment 1: In this revised manuscript, the authors have appropriately addressed the concerns raised by me and other reviewers. I recommend the manuscript for publication in Nature Communications.

Our response: We appreciate the Reviewer's thorough assessment and recommendation for publication.

Comment 2: Typos: Figure 9: the subtitle in both the text and figure should read "metallaphotocatalytic" instead of "metallophotocatalytic."

Our response: We thank the Reviewer for catching this typographical error. We have corrected "metallophotocatalytic" to "metallaphotocatalytic" in the Figure 9 subtitle.